# Research on robust adaptive control of strong nonlinear complex large power grids

Xiaoping Huang[1]*, Yongji Chen[2], Yaqiong Zhang[1], Pengcheng Wang[1], Wenzhe Huang[3]

**1** Guilin University of Technology, Guilin, Guangxi, China, **2** Nanning Elevator Industry Associate, Nanning, Guangxi, China, **3** Shanghai University of Electric Power, Shanghai, Shanghai, China

* 9821061@glut.edu.cn

## Abstract

In this paper, a Multi-Index Nonlinear Robust Adaptive Control (MINRAC) method was proposed for ultra-complex multi-input multi-output power systems with uncertain parameter factors and external disturbances. The controller designed by this method has excellent static and dynamic characteristics. Under the condition of the uncertainty of system parameters and external disturbance at the same time, the MINRAC method can ensure that the multiple indexes concerned by ultra-complex power systems can be controlled at their expected values. The simulation results showed that the control mechanism of MINRAC method was consistent in both single-machine infinite bus system and multi-machine interconnected coupling system. The output function chooses power angle, angular frequency and terminal voltage as constraints. When the system has parameter uncertainty and external interference, the uncertain parameter values are adjusted by adaptive control to force these indicators to tend to the given expected value. For three-phase short circuit, which is the most serious fault in power system, the use of multi index nonlinear robust controller can ensure that the system is stable in a wide range and has better dynamic performance.

## 1 Introduction

The grid form of the new power system shows the integration of UHV main grid, micro grid and local area network, and the coexistence of AC power grid and AC and DC distribution network. With the construction of new power systems and new energy power with huge randomness and volatility entering the power grid, how to achieve power balance control within the ultra-complex power grid and optimize system operation plays an important role in improving the power supply reliability of the power system. For the control research of complex large power grids, scholars have achieved fruitful results by combining backstep method with passivity theory, adaptive theory and robust control theory of the system, but there are still some areas that need to be improved. For example: The inverse step method is mainly used in a class of nonlinear control systems with lower triangular structure, and for some complex large power grids such as flexible alternating current transmission systems and generator coordination control system [1–3], the differential algebraic model is complex and of high order. For backstep method, each order corresponds to a virtual controller. The higher the order, the more complicated the controller design process [4,5].

**Data Availability Statement:** All data generated or analyzed during this study are included in the paper.

**Funding:** This work was supported by Science and Technology Project of Chongzuo, No.

Chongke20231206. The funder had no role in study design, data collection and analysis, decision to publish, or preparation of the manuscript.

**Competing interests:** The authors have declared that no competing interests exist.

**Abbreviations:** *MINRAC*, Multi-Index Nonlinear Robust Adaptive Control; *MINRC*, Multi-index nonlinear robust control.

However, the differential geometry method is not limited by the order of the system, and has a good application in many complex and high-order nonlinear control systems [6,7]. However, the essence of the differential geometry method is the accurate cancellation of nonlinear terms, which has high requirements for the accuracy of the system. Therefore, when scholars use differential geometry to design controllers, they almost never take into account the uncertainty factors existing in the system. Only a small number of scholars have conducted research in this area. One of the most representative is the robust control of uncertain power systems based on accurate feedback linearization, but this method requires the output function of the system to be the same as the order of the system, so it is difficult to select the output function [8,9]. Moreover, the uncertain power system model only considers the external interference of the system, without considering the parameter uncertainty of the system. The output function of multi-index nonlinear control is easy to select and can stably control multiple indicators of the system, and the zero poles of the system can be arbitrarily configured by setting the coefficient matrix of the output function. However, current studies are based on accurate systems or only external interference, and do not take into account the uncertainty of system parameters [10–13]. To solve the above problems, this paper applies multi-index nonlinear design to uncertain power systems with uncertain parameters and external interference, combines it with robust control theory and adaptive control theory, proposes a multiindex nonlinear robust adaptive control method, and gives its closed-loop stability proof [14,15]. In [38,39], a decentralized controller based on Lyapunov theory for excitation control of multi-machine power systems ensured global asymptotic stability and realized voltage regulation through excitation control. In [40,41], For the multi-electromechanical power system including large-scale photovoltaic energy storage power station, the multi-objective holographic feedback control method is adopted. Through the test of the IEEE 6 machine 24-node system, the control method can effectively suppress the voltage and frequency oscillation under the most serious fault of the power system during the three-phase short circuit, and can quickly recover to the stable rated operating state after the short circuit is removed. The controller designed by this method has excellent dynamic and static control performance. In the case of parameter uncertainty and external disturbance in the system, the controller can ensure that the multiple indicators concerned in the system can be controlled at their expected values without static deviation. For the dynamic performance of the system, it is stable in a large range and the response speed is fast, and the control process is smooth. MINRAC can also adapt to the change of network parameters for the change of power network structure(such as network disconnection fault), and its control effect is obviously better than other control methods.

## 2 Differential algebraic model of multi-machine power systems

Consider a multi-machine interconnected power system with $m$ generators and m nodes, all m generators adopt the third-order classical model, and the differential equation of the multi-machine power system can be obtained as follows:

$$\begin{cases} \dot{\delta}_i = \omega_0(\omega_i - 1) \\ \dot{\omega}_i = \dfrac{1}{T_{ji}}(P_{mi} - P_{ei}) - \dfrac{D_i}{T_{ji}}(\omega_i - 1)(i = 1, 2, \cdots m) \\ \dot{E}'qi = -\dfrac{1}{T'_{doi}}E_{qi} + \dfrac{K_i}{T'_{doi}}U_i + d_i \end{cases} \tag{1}$$

Parameters in Eq (1) were described as follows.
$\omega_0$ -Synchronous angular velocity;

$\Delta_i$ -Power angle;

$\omega_j$ -Angular frequency;

$P_{mi}$ -Input mechanical power of the $i$ generator;

$P_{ei}$ -Output active power of the $i$ generator;

$E_{qi}$ -No-load potential of the $i$ generator;

$E'_{qi}$ -Transient potential of the $i$ generator;

$u_i$ -Excitation voltage of the $i$ generator;

$D_i$ -Damping winding coefficient of the $i$ generator;

$T_{ji}$ -Inertia time constant of the generator rotor;

$T_{d0i}$ -Damping winding time constant when the stator is open;

$K_i$ -Error coefficient between the actual and ideal input of the controller;

$di$ -External electromagnetic interference of the generator.

Load adopts a constant impedance model, and load impedance was obtained according to the active power, reactive power and reference power is Eq (2)[1–4].

$$Z = \frac{V_N^2}{V_S} = \frac{V_N^2}{P_N + jQ_N} = \frac{V_N^2(P_N - jQ_N)}{P_N^2 + Q_N^2}$$
$$= \frac{V_N^2(P_N - jQ_N)}{S_N^2} \tag{2}$$

Taking the reference power $S_B$, and reference voltage $V_B = V_N$, the unit value of load impedance can be obtained in Eq (3).

$$Z^* = \frac{1}{S_N^{*2}}(P_N^* - jQ_N^*) \tag{3}$$

Then the per-unit value of load admittance is Eq (4).

$$Y^* = P^* + jQ_N^* \tag{4}$$

According to the power flow equation of the network in Eq (5)[5,6].

$$\begin{bmatrix} I_g \\ 0 \end{bmatrix} = \begin{bmatrix} Y_{gg} & Y_{gL} \\ Y_{Lg} & Y_{LL} \end{bmatrix} \begin{bmatrix} E'_q \\ V_L \end{bmatrix} \tag{5}$$

In Eq (5), $I_g$ represents the current vector of synchronous generator; $Y_{gg}, Y_{gL}, Y_{LL}$ are respectively represent the self-admittance matrix of synchronous generator, the mutual admittance matrix between synchronous generator and load nodes, and the self-admittance matrix between load nodes; $E'_q$, $V_L$ are respectively represent voltage vectors of synchronous generators and load nodes.

According to the electrical network theory, all the passive nodes are eliminated, and only the generator nodes of the system are retained, and the $n$-order system is reduced to the $m$-order system, as shown in Fig 1.

After order reduction, the current of each generator in the grid is Eq (6).

$$I_g = (Y_{gg} - Y_{gL}Y_{LL}^{-1}Y_{Lg})E'_q = Y'_{gg}E'_q \tag{6}$$

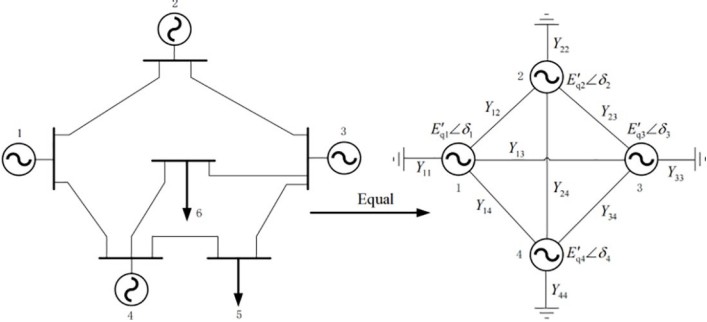

**Fig 1. Reduced order equivalent diagram of multi machine power system.**

The complex power of each generator is Eq (7).

$$
\begin{aligned}
S_i &= \dot{E}'_{qi} I_i \\
&= E'_{qi} \sum_{j=1}^{m} E'_{qj} (\cos \delta_{ij} + j \sin \delta_{ij})(G_{ij} - jB_{ij}) \\
&= E'_{qi} \sum_{j=1}^{m} E'_{qj} (G_{ij} \cos \delta_{ij} + B_{ij} \sin \delta_{ij}) + jE'_{qi} \sum_{j=1}^{m} E'_{qj} \cdot (G_{ij} \sin \delta_{ij} - B_{ij} \cos \delta_{ij})
\end{aligned}
\tag{7}
$$

In Eq (7):

$\delta_{ij} = \delta_i$-$\delta_j$ denotes the work angle difference between the $i$ generator and the $j$ generator;

$Y'_{gg} = G + jB$, $G_{ij}$, $B_{i}j$ denote the real and imaginary parts of the reduced order node admittance matrix respectively.

According to Eq (7), the current $I_i$ of the $i$ generator, its direct axis component $I_{di}$ and its quadratic axis component $I_{qi}$ are Eq (8)[7–9].

$$
\begin{cases}
I_{qi} = \sum_{j=1}^{m} E'_{qj} (G_{ij} \cos \delta_{ij} + B_{ij} \sin \delta_{ij}) \\
I_{di} = \sum_{j=1}^{m} E'_{qj} (G_{ij} \sin \delta_{ij} - B_{ij} \cos \delta_{ij}) \\
I_i = \sqrt{I^2_{di} + I^2_{qi}}
\end{cases}
\tag{8}
$$

The power voltage and no-load potential constraints of the $i$ generator is Eq (9).

$$
\begin{cases}
P_{ei} = E'_{qi} I_{qi} \\
E_{qi} = E'_{qi} + (x_{di} - x'_{di}) I_{di} \\
U_{gdi} = x'_{di} I_{qi} \\
U_{gqi} = E'_{qi} - x'_{di} I_{di} \\
U_{gi} = \sqrt{U^2_{gdi} + U^2_{gqi}}
\end{cases}
\tag{9}
$$

In Eq (9): $U_{gi}$, $U_{gdi}$, $U_{gqi}$ were respectively the terminal voltage of the $i$ generator and its direct axis and quadrature axis components; $x_{di}$, $x'_{di}$ were respectively the direct axis reactance and the direct axis transient reactance of the $i$ generator[11–13].

Considered the damping coefficient $D_i$ of each generator in the multi-machine power system, the damping winding time constant $T''_{doi}$ and the excitation input error coefficient $K_i$ when the stator was open are uncertain parameters of the system, and the external electromagnetic interference $d_i$ existed in the system.

Then $\theta$ can be assumed as Eq (10).

$$\theta = [\theta_{1i}, \theta_{2i}, \theta_{3i}]^T = \left[D_i/T_{Ji}, 1/T'_{doi}, K_i/T'_{doi}\right]^T \tag{10}$$

The output function of the system was shown in Eq (11).

$$y_i = c_{1i}\Delta U_{gi} + c_{2i}\Delta\omega_i \tag{11}$$

According to Eqs (1), (8) and (9), the differential algebraic equation of the $i$ generator can be written in Eq (12) [14,15].

$$\begin{cases} \dot{x} = f(x, w, \theta) + G_1(x, \omega, \theta)u + G_2(x, \omega)d \\ \omega = p(x, \omega) \\ y = h(x, \omega) \end{cases} \tag{12}$$

In Eq (12): $x, \omega$ must accord with Eq (13).

$$\begin{cases} x = [x_{1i}, x_{2i}, x_{3i}]^T = \left[\delta_i, \omega_i - 1, E'_{qi}\right]^T \\ \omega = [\omega_{1i}, \omega_{2i}, \omega_{3i}, \omega_{4i}, \omega_{5i}]^T = \left[P_{ei}, E_{qi}, U_{gi}, I_{di}, I_{qi}\right]^T \end{cases} \tag{13}$$

$f(x, \omega, \theta)$ must conform to Eq (14).

$$f(x, \omega, \theta) = \begin{bmatrix} \omega_0(\omega_i - 1) \\ \dfrac{1}{T_{Ji}}(P_{mi} - P_{ei}) - \theta_{1i}(\omega_i - 1) \\ -\theta_{2i}E_{qi} \end{bmatrix} \tag{14}$$

$G_1(x, \omega, \theta), G_2(x, \omega)$ must conform to Eq (15).

$$\begin{cases} G_1(x, \omega, \theta) = [g_1] = \begin{bmatrix} 0 \\ 0 \\ \theta_{3i} \end{bmatrix} \\ G_2(x, \omega) = [g_2] = \begin{bmatrix} 0 \\ 0 \\ 1 \end{bmatrix} \end{cases} \tag{15}$$

$p(x,\omega)$ must conform to Eq (16).

$$p(x,\omega) = \begin{bmatrix} E'_{qi}I_{qi} - P_{ei} \\ E'_{qi} + (x_{di} - x'_{di}) - E_{qi} \\ \sqrt{(x'_{di}I_{qi})^2 + (E'_{qi} - x'_{di}I_{di})^2} - U_{gi} \\ \sum_{j=1}^{m} E'_{qi}(G_{ij}\sin\delta_{ij} - B_{ij}\cos\delta_{ij}) - I_{di} \\ \sum_{j=1}^{m} E'_{qi}(G_{ij}\cos\delta_{ij} + B_{ij}\sin\delta_{ij}) - I_{qi} \end{bmatrix} \quad (16)$$

$x,d,y$ must accord with Eq (17).

$$\begin{cases} u = u_i \\ d = d_i \\ y = h(x,\omega) = \begin{bmatrix} c_{1i}\Delta U_{gi} + c_{2i}\Delta\omega_i \end{bmatrix} \end{cases} \quad (17)$$

## 3 Minrac controller design of multi-machine power system

According to the multi-index nonlinear control theory, first calculate the matrix sum $\frac{\partial p}{\partial x}$ and $\frac{\partial p}{\partial \omega}$ [16–19].

$$\begin{cases} \frac{\partial p}{\partial x} = \begin{bmatrix} 0 & 0 & x_{13} \\ 0 & 0 & x_{23} \\ 0 & 0 & x_{33} \\ x_{41} & 0 & x_{43} \\ x_{51} & 0 & x_{53} \end{bmatrix} \\ \frac{\partial p}{\partial \omega} = \begin{bmatrix} -1 & 0 & 0 & 0 & \sigma_{15} \\ 0 & -1 & 0 & \sigma_{24} & 0 \\ 0 & 0 & -1 & \sigma_{34} & \sigma_{35} \\ 0 & 0 & 0 & -1 & 0 \\ 0 & 0 & 0 & 0 & -1 \end{bmatrix} \end{cases} \quad (18)$$

Each element in the $\frac{\partial p}{\partial x}$ matrix is Eq (19) [20–23].

$$
\begin{cases}
x_{13} = I_{qi} \\
x_{23} = 1 \\
x_{33} = \dfrac{E'_{qi} - x'_{di}I_{di}}{U_{gi}} \\
x_{43} = -B_{ii} \\
x_{53} = G_{ii} \\
x_{41} = \displaystyle\sum_{j=1,j\neq i}^{m} E'_{qi}(G_{ij}\cos\delta_{ij} + B_{ij}\sin\delta_{ij}) \\
x_{51} = \displaystyle\sum_{j=1,j\neq i}^{m} E'_{qi}(-G_{ij}\sin\delta_{ij} + B_{ij}\cos\delta_{ij})
\end{cases}
\tag{19}
$$

Each element in the $\frac{\partial p}{\partial \omega}$ matrix is Eq (20).

$$
\begin{cases}
\sigma_{15} = E'_{qi} \\
\sigma_{24} = x_{di} - x'_{di} \\
\sigma_{34} = \dfrac{-(E'_{qi} - x'_{di}I_d i)x'_{di}}{U_{gi}} \\
\sigma_{35} = \dfrac{x'^2_{di}I_{qi}}{U_{gi}}
\end{cases}
\tag{20}
$$

There are:

$$
\begin{cases}
M_{g1}h &= \theta_{3i}[c_{2i}(x_{33} + \sigma_{34}x_{43} + \sigma_{35}x_{53})] \\
M_f h &= \theta_{1i}[-c_{1i}(\omega_i - 1)] + \theta_{2i}[-E_{qi}c_{2i}(x_{33} + \sigma_{34}x_{43} + \sigma_{35}x_{53})] \\
&\quad + c_{1i}\dfrac{P_{mi} - P_{ei}}{T_{Ji}} + c_{2i}\omega_0(\omega_i - 1)(\sigma_{34}x_{41} + \sigma_{35}x_{51}) \\
&= \theta_{1i}\varphi_{1i} + \theta_{2i}\varphi_{2i} + \bar{\varphi}_i
\end{cases}
\tag{21}
$$

The multi-index nonlinear robust controller of multi-machine power system is Eq (22) [24–27].

$$
u_i = \frac{v_i - M_f h_i}{M_{g1}h_i}
\tag{22}
$$

Here is: $v_i = -k_i z_i = -k_i y_i$.

According to the design steps of adaptive control, the input $v_{i*}$ of the subsystem meets Eq (23) [28–32].

$$
\begin{aligned}
v_{i*} &= M_f h_i(x, \omega, \theta) + M_{g1}h_i(x, \omega, \theta)\tilde{u}_i \\
&= v_i + M_f h_i(x, \omega, \theta) - M_f h_i(x, \omega, \tilde{\theta}) + \left[M_{g1}h_i(x, \omega, \theta) - M_{g1}h_i(x, \omega, \tilde{\theta})\right]\tilde{u}_i \\
&= vi + (\theta_{1i} - \tilde{\theta}_{1i})\varphi_{1i} + (\theta_{2i} - \tilde{\theta}_{2i})\varphi_{2i} + (\theta_{3i} - \tilde{\theta}_{3i})[c_{2i}(x_{33} + \sigma_{34}x_{43} + \sigma_{35}x_{53})]\tilde{u}_i
\end{aligned}
\tag{23}
$$

There are:

$$\begin{cases} W_{1i} = \varphi_{1i} \\ W_{2i} = \varphi_{2i} \\ W_{3i} = \varphi_{3i} = [c_{2i}(x_{33} + \sigma_{34}x_{43} + \sigma_{35}x_{53})]\tilde{u}_i \end{cases} \tag{24}$$

The extended Lyapunov function of subsystem $Z_i$ is chosen as follows in Eq (25) [33–35].

$$V(z_i, \tilde{\theta}) = \frac{1}{2}z_i^2 + \sum_{j=1}^{3} \frac{1}{2\lambda_{ji}}(\tilde{\theta}_{ji} - \theta_{ji})^2 \tag{25}$$

$\lambda_{ji}$ represents the adaptive rate of the j uncertain parameter of the $i$ generator in Eq (26) [36–38].

$$\frac{\mathrm{d}V}{dt} = z_i v_{i*} + \sum_{j=1}^{3} \frac{1}{\lambda_{ji}}(\tilde{\theta}_{ji} - \theta_{ji})\dot{\tilde{\theta}}_{ji} \qquad = z_i v_i - \sum_{j=1}^{3}(\tilde{\theta}_{ji} - \theta_{ji})z_i W_{ji} + \sum_{j=1}^{3} \frac{1}{\lambda_{ji}}(\tilde{\theta}_{ji} - \theta_{ji})\dot{\tilde{\theta}}_{ji} \tag{26}$$

We can assign:

$$\dot{\tilde{\theta}}_{ji} = \lambda_{ji}W_{ji}z_i(j = 1, 2, 3) \tag{27}$$

Then Eq (23) was converted to Eq (28).

$$\frac{dV}{dt} = z_i v_i = -k_i z_i^2 \leq 0 \tag{28}$$

In this way, the MINRAC lawfor the multi-machine power system is Eq (29) [39–41].

$$\begin{cases} u_i = \dfrac{-k_i z_i - M_f h_i(x, \omega, \tilde{\theta})}{M_{g1} h_i(x, \omega, \tilde{\theta})} \\ \theta_{ji} = \begin{cases} \lambda_{ji} W_{ji} z_i(j = 1, 2, 3)\|z_i\|^2 < u_i \\ \qquad\qquad\qquad\qquad\qquad (u_i > 0) \\ 0\|z_i\|^2 \geq u_i \end{cases} \end{cases} \tag{29}$$

## 4 IEEE 3-machine 9-node system

### 4.1 Structure and system parameters of IEEE 3-machine 9-node system

In order to verify the effectiveness of MINRAC control law, the control law was applied to IEEE 3-machine 9-nodesystem. Usually, IEEE 3-machine 9-node system was a classic power system model, consisting of 9 nodes and 12 branches. The system was characterized by its moderate size and the inclusion of common power system components, so it was often used in academic research and engineering practice, and it was shown in Fig 2.

In Fig 2, G1 was set as a balanced node that did not participate in the control of the system. Both G2 and G3 were equipped with MINRAC controllers, and G2 and G3 were also equipped with MINRC controllers for comparison.

The data of line, generator and load were shown in Tables 1–4, and the unit of parameters in the table was the standard unit value.

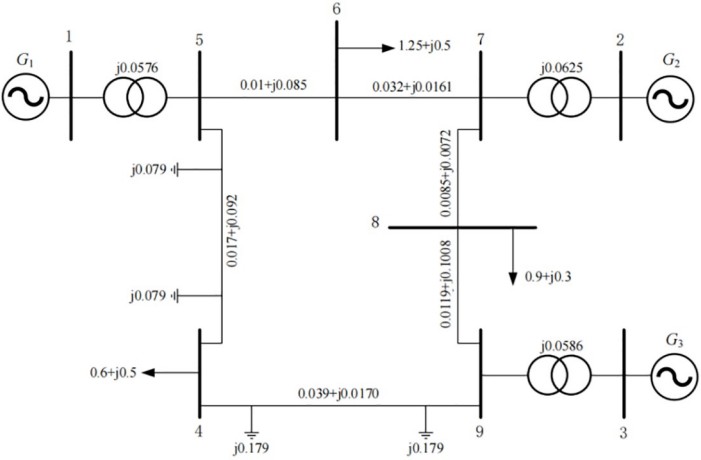

**Fig 2. Three-machines nine-bus power system.**

All passive nodes of the system are eliminated (that is, only generator nodes 1, 2 and 3 are retained), and the node admittance moment of the system after order reduction is Eq (30).

$$Y'_{gg} = G + jB$$

$$= \begin{bmatrix} 1.176 & -0.342 & -0.276 \\ -0.342 & 1.267 & 0.079 \\ -0.276 & 0.079 & 0.618 \end{bmatrix} + j \begin{bmatrix} -4.499 & 3.565 & 1.307 \\ 3.565 & -6.651 & 3.518 \\ 1.307 & 3.518 & -4.533 \end{bmatrix} \tag{30}$$

## 4.2 Configuring controller parameters

For the MINRC controller configuration:

Machine 2: $c_{21} = 200$, $c_{22} = 2.4$, $k_2 = 5$;

Machine 3: $c_{31} = 150$, $c_{32} = 2.5$, $k_3 = 8.5$.

Compared with the MINRAC controller, the multi-index nonlinear robust control part of the MINRAC controller remains unchanged, and the adaptive controller is more.

Therefore, MINRAC controller parameters $c$ and $k$ remain unchanged. For the adaptive part:

**Table 1. Line parameters.**

| No. | First node | End point indicator | Resistor(R) | Reactance(X) | Susceptance(B/2) |
| --- | --- | --- | --- | --- | --- |
| 1 | 4 | 5 | 0.017 | 0.092 | 0.079 |
| 2 | 4 | 9 | 0.039 | 0.017 | 0.179 |
| 3 | 5 | 6 | 0.01 | 0.085 | 0.000 |
| 4 | 6 | 7 | 0.032 | 0.0161 | 0.000 |
| 5 | 7 | 8 | 0.0085 | 0.0072 | 0.000 |
| 6 | 8 | 9 | 0.0119 | 0.1008 | 0.000 |

**Table 2. Transformer parameters.**

| No. | First node | End point indicator | Resistor (R) | Reactance (X) | Variable-ratio |
|---|---|---|---|---|---|
| 1 | 1 | 5 | 0 | 0.0576 | 1.0 |
| 2 | 2 | 7 | 0 | 0.0625 | 1.0 |
| 3 | 3 | 9 | 0 | 0.0586 | 1.0 |

**Table 3. Load power.**

| No. | Generatrix | Active power (P) | Reactive power (Q) |
|---|---|---|---|
| 1 | 4 | 0.6 | 0.5 |
| 2 | 6 | 1.25 | 0.5 |
| 3 | 8 | 0.9 | 0.3 |

**Table 4. Generator parameters.**

| No. | Generator | $x_d$ | $x'_d$ | D | TJ | $T'_{d0}$ | K |
|---|---|---|---|---|---|---|---|
| 1 | 2 | 0.77 | 1.121 | 15.62 | 3.0 | 6.2 | 1 |
| 2 | 3 | 1.3125 | 0.1813 | 6.02 | 3.0 | 5.8 | 1 |

Machine 2: $\lambda_{21} = 500$, $\lambda_{22} = 10$, $\lambda_{23} = 0.1$, $\mu_2 = 2$;

Machine 3: $\lambda_{31} = 550$, $\lambda_{32} = 5$, $\lambda_{33} = 1$, $\mu_3 = 2$.

The initial operating condition of the system:

Machine 2: $\delta_{20} = 0.2181$rad, $U_{g20} = 1.025pu$, $P_{e20} = 0.85p$u;

Machine 3: $\delta_{30} = 0.541$rad, $U_{g30} = 1.025pu$, $P_{e30} = 1.63pu$.

The transient potential of machine 1 was also given $E'_{q1} = 0.9547pu$.

## 5 Analysis of simulation result

### 5.1 Anti-external interference performance and voltage regulation performance of the system

The system was stable in the first 5s and the system parameters were accurate, that was $\theta = \tilde{\theta}$, external interference occurs in the system when $t = 5$s, disturbance $d = [d_2, d_3]^T = [0.2, 0.2]^T$, since then the disturbance had been there. When $t = 10$s, $G_2$ terminal voltage was increased by 2.5%, and $G_3$ terminal voltage was decreased by 2.5%. In this way, the anti-external interference ability and excitation voltage regulation performance of the system were investigated.

As shown in Fig 3, the system was not affected by external disturbance in the initial 5s, and the system could maintain stable operation under the action of MINRAC controller. When there was an external disturbance in the system $t = 5$s and the disturbance persists thereafter: Fig 3(a) showed that the MINRC controller could be robustly stabilized quickly, and the power angles of $G_2$ and $G_3$ would be stabilized again within 2s. The stabilization time of MINRAC was slightly longer than that of MINRC controller according to the different adaptive rates.

However, it could be seen from Fig 3(c) that under the control of MINRC controller, due to the influence of external disturbance, the terminal voltage of $G_2$ decreases by about 2.5% and

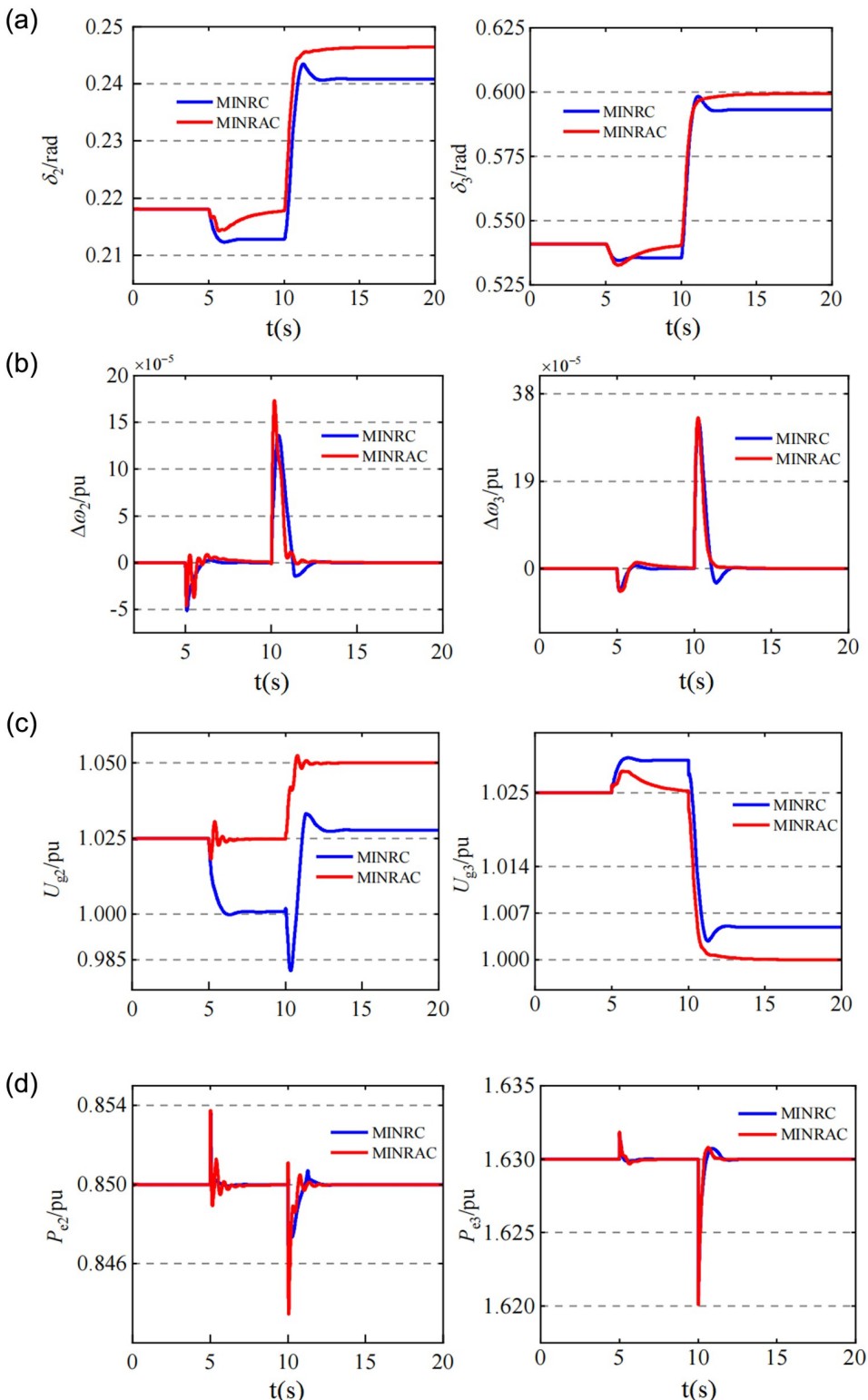

**Fig 3. External interference occurs in 5s, and the given value of $G_2$ and $G_3$ voltage changes, the change curve of relevant state quantity of the system in 10s.** (a) The work angle change curves of $G_2$ and $G_3$. (b) Angular frequency change curves of $G_2$ and $G_3$. (c) The terminal voltage change curves of $G_2$ and $G_3$. (d) Change curves of output active power of G2 and G3.

the terminal voltage of $G_3$ rised by about 0.5%, both deviating from their original working conditions.

Under the control of MINRAC, the terminal voltage of $G_2$ and $G_3$ were maintained at the original given value without deviation, and other indicators of the system, such as power angle, angular frequency and output active power, which were also maintained at the original given value, indicated that the MINRAC controller also had excellent anti-external interference ability in the multi-machine interconnected power system.

When $t$ = 10s, the system balance point changes, the terminal voltage $U_{g20}$ of $G_2$ was increased by 2.5% to 1.05$Pu$, and the terminal voltage $U_{g30}$ of $G_3$ is decreased by 2.5% to 1.0$Pu$. Therefore, the voltage regulation performance of the controller was considered.

It can be seen from Fig 3(c) that under the control of MINRAC controller, the terminal voltage of G2 and G3 can quickly track the given value, while under the MINRC method, the terminal voltage of G2 and G3 tended to be close to the given value due to the constant presence of external interference, but there was always a steady-state error with the given value. It can be seen from Fig 3(a) that the power angles of G2 and G3 change to adapt to the new equilibrium point due to the change of terminal voltage.

## 5.2 System disconnection fault

When t = 5s, a permanent disconnection fault occurs in the primary line during the double-circuit line operation between lines 6~7. That was, the system network parameters were changed. Therefore, the adaptability of MINRAC controller to network parameters in the multi-machine interconnected power system was investigated.

From Fig 4, we can identify the external disturbance, and compensate the disturbance by adjusting parameters through adaptive control, which can maintain the stability of the terminal voltage without generating steady-state offset.

To highlight the control effect in this case, the external disturbance of the system was set to d = [d2, d3] T = [0, 0]T.

Fig 5 showed that both MINRC and MINRAC could maintain the terminal voltage and output active power of the generator at the original given value. Due to the change of network parameters, the power angle of the motor was bound to change to adapt to the new equilibrium point when the terminal voltage and active power remained unchanged.

Fig 5(a) was in line with the normal physical process of the generator. When t = 5s, there was a line break fault.

Fig 5(d) showed that when faults occur, the voltage overshoot of G2 and G3 terminals under the action of MINRAC controller was less than 2.5%, while the voltage overshoot of G2 terminals under the action of MINRC controller even exceeds 20%, and there was a drastic oscillation process, indicated that MINRAC controller had a better ability to maintain the voltage stability of generator terminals.

Fig 5(e) showed that MINRAC controller could identify external disturbances and quickly adapt to changes in network parameters by adjusting uncertain parameters in a short time. Besides, the adjustment process of power angle of generator was smooth, and there was no overshoot compared with MINRC controller.

It is worth mentioning that the uncertainty of network parameters in the algebraic equations is not considered in the derivation of MINRAC controller. However, from the simulation results, MINRAC can also identify the external disturbance, and compensate the disturbance by adjusting the uncertain parameters through adaptive control, which can maintain the stability of the terminal voltage without generating steady-state offset. The adjustment process is fast and smooth, and its control effect is significantly better than MINRC.

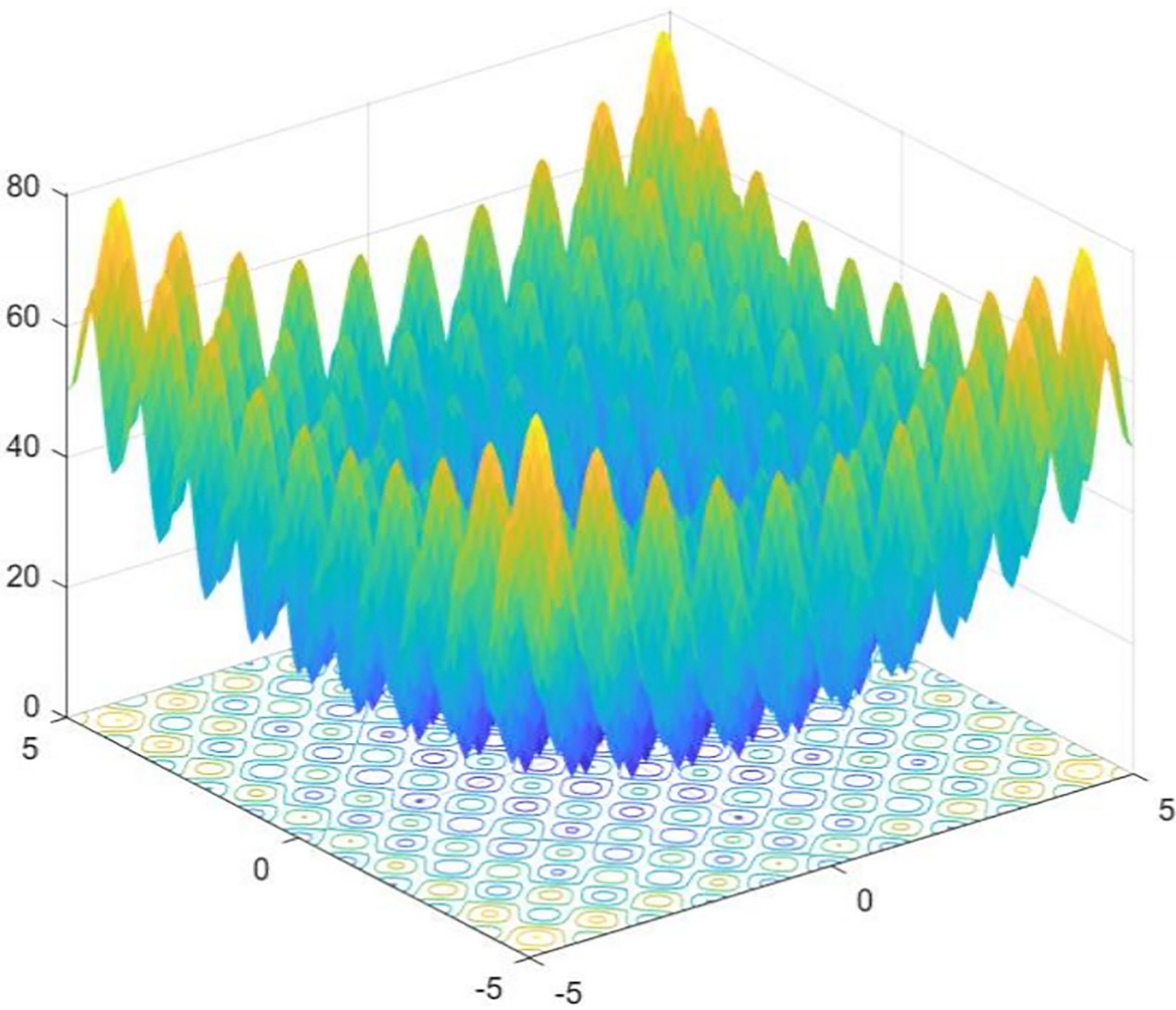

**Fig 4. Stability control diagram of terminal voltage.**

## 6 Conclusion

By establishing a differential algebraic model of multi-machine power system with uncertain factors, the differential equation was composed of the rotor motion equation and excitation part of each generator in the network, and the electronic current of each generator in the algebraic equation was not only dependent on itself, but also related to the work angle, transient potential and network parameters of other generators. This was also the difference between the multi-machine power system and the single-machine infinite system. The MINRAC controller was designed and tested by IEEE 3-machine 9-node system. The conclusions were summarized as follows:

- In multi-machine power system, for the uncertainty of the generator itself, such as damping coefficient, rotor inertia time constant is inaccurate and generator excitation of electromagnetic disturbance, MINRAC method effect rather than in a single machine infinite system, the mechanism of selection by the output function system of the relationship between

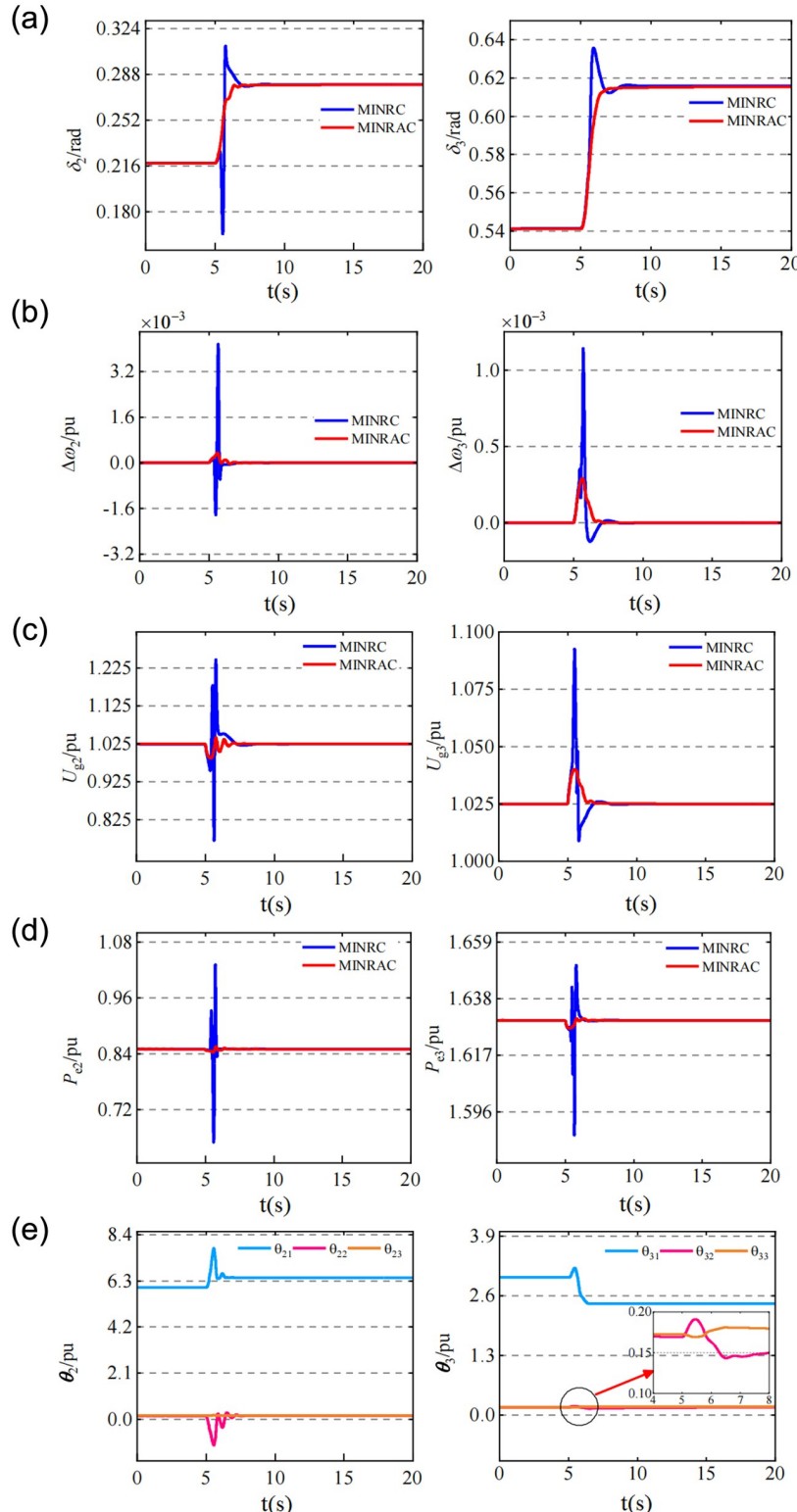

**Fig 5. Change curves of relevant state variables and uncertain parameters of the system under the action of two controllers when the system has a disconnection fault.** (a) The work angle change curves of $G_2$ and $G_3$. (b) Angular frequency change curves of $G_2$ and $G_3$. (c) The terminal voltage change curves of $G_2$ and $G_3$. (d) Change curves of output active power of $G_2$ and $G_3$. (e) Change curves of uncertain parameters in $G_2$ and $G_3$.

indicators such as angle, angular frequency and voltage of the machine as a constraint. By adjusting the uncertain parameter value by adaptive control, these indexes are forced to tend to the given expected value, so as to achieve static tracking.

- For the uncertainty of network parameters (such as permanent line break fault in the primary line of double loop), although this was not considered in the derivation of MINRAC method, from the simulation results, MINRAC can also identify the changes of network parameters and greatly reduce rotor sway through adaptive control. The overshoot of generator terminal voltage is also much less than that of MINRC method, which further improves the robustness of the system.

Futhermore, for power systems with time delay, how to optimize the adaptive control law to ensure the stability of the system when there are irregular signals with high signal amplitude needs to be verified by further engineering practice.

## Author Contributions

**Conceptualization:** Xiaoping Huang.

**Data curation:** Yongji Chen, Wenzhe Huang.

**Formal analysis:** Wenzhe Huang.

**Funding acquisition:** Xiaoping Huang.

**Investigation:** Yaqiong Zhang.

**Methodology:** Xiaoping Huang.

**Software:** Yongji Chen, Pengcheng Wang.

**Supervision:** Yongji Chen, Pengcheng Wang.

**Writing – original draft:** Xiaoping Huang, Yaqiong Zhang, Wenzhe Huang.

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
