## [Decision Letter · Decision Letter 0]

9 Feb 2024

PONE-D-23-43214Research on Robust Adaptive Control of Strong Nonlinear Complex Large Power GridsPLOS ONE

Dear Dr. Huang,

Thank you for submitting your manuscript to PLOS ONE. After careful consideration, we feel that it has merit but does not fully meet PLOS ONE’s publication criteria as it currently stands. Therefore, we invite you to submit a revised version of the manuscript that addresses the points raised during the review process.

**ACADEMIC EDITOR: **The reviewers recommend reconsideration the manuscript with revision and modification. I invite the authors to resubmit the manuscript after addressing the comments raised by the reviewers.

We look forward to receiving your revised manuscript.

Kind regards,

Dhanamjayulu C, Ph.D & Post.Doc

Academic Editor

PLOS ONE

“This work was supported by Science and Technology Project of Chongzui,No.Chongke20231206.”

“This work was supported by Science and Technology Project of Chongzui,No.Chongke20231206.”

“This work was supported by Science and Technology Project of Chongzui,No.Chongke20231206.”

5. Please keep your tables as part of your main manuscript and remove the individual files. Please note that supplementary tables (should remain/ be uploaded) as separate "supporting information" files.

Additional Editor Comments:

The reviewers recommend reconsideration the manuscript with revision and modification. I invite the authors to resubmit the manuscript after addressing the comments raised by the reviewers.

Reviewers' comments:

Reviewer's Responses to Questions

**Comments to the Author**

1. Is the manuscript technically sound, and do the data support the conclusions?

Reviewer #1: Yes

Reviewer #2: Yes

2. Has the statistical analysis been performed appropriately and rigorously? 

Reviewer #1: Yes

Reviewer #2: Yes

3. Have the authors made all data underlying the findings in their manuscript fully available?

Reviewer #1: Yes

Reviewer #2: Yes

4. Is the manuscript presented in an intelligible fashion and written in standard English?

Reviewer #1: Yes

Reviewer #2: Yes

5. Review Comments to the Author

Reviewer #1: This paper proposes a multi-index nonlinear robust adaptive control method for ultra-complex multi-input multi-output power systems with uncertain parameter factors and external disturbances. Overall, the research motivation of this paper has a specific positive significance. Before publication, some necessary corrections should be carried out and the detailed comments are listed below.

1- What is an "ultra-complex multiple-input multiple-output power system"? Please explain in detail in the text.

2- To make it easier for the reader to understand, please explain the variables that appear for the first time in the paper. In addition, there are some errors in the symbols in the paper. It is recommended to check the symbols throughout the paper.

3- What does the MINRC in section IV IEEE 3-MACHINE 9-NODE SYSTEM stand for? To avoid confusing the reader, please explain it clearly in the paper.

4- The authors claim that the proposed control method can ensure that the multiple indexes concerned by ultra-complex power systems can be controlled at their expected values under the condition of the uncertainty of system parameters. However, the authors said in the simulation that the system parameters are accurate. This is confusing operation, please make a detailed explanation.

5- The authors should add some more recent relevant references in the references section and cite them properly in the paper.

6- It is recommended that the author reformulate an appropriate title based on the content of the paper.

7- In conclusion part, more future works and challenges should be recommended.

8- To improve the quality of the paper, it is recommended to improve the English writing skills.

Reviewer #2: In this work, a Multi-Index Nonlinear Robust Adaptive Control (MINRAC) method has been proposed for ultra-complex multi-input multi-output power systems with uncertain parameter factors and external disturbances. However, in order to further improve the quality of this manuscript, the comments in the uploaded file should be carefully considered.

6. PLOS authors have the option to publish the peer review history of their article (what does this mean?). If published, this will include your full peer review and any attached files.

Reviewer #1: No

Reviewer #2: No

---

## [Author Response · Author response to Decision Letter 0]

17 Feb 2024

Comments to the authors of academic editor: 

 The reviewers recommend reconsideration the manuscript with revision and modification.I invite the authors to resubmit the manuscript after addressing the comments raised by the reviewers.

 Reply: Many thanks for your kind handling of the review of our paper. Following your suggestions,we have carefully revised the paper by clarifying the given questions as explained below. 

Comments to the authors of Reviewer #1:

 1. What is an "ultra-complex multiple-input multiple-output power system"? Please explain in detail in the text.

 Reply: "Ultra-complex multiple-input multiple-output power system" mainly refers to the smart grid system that can connect the major coal power base, the major hydropower base, the major nuclear power base and the major renewable energy base.And the content had been added in “1 INTRODUCTION” on Page 1.

 2. To make it easier for the reader to understand, please explain the variables that appear for the first time in the paper. In addition, there are some errors in the symbols in the paper. It is recommended to check the symbols throughout the paper.

 Reply: We have added the variables that appear for the first time in the paper in “NOMENCLATURE” on Page 1,and corredted the errors of symbols in the paper. 

 3. What does the MINRC in section IV IEEE 3-MACHINE 9-NODE SYSTEM stand for? To avoid confusing the reader, please explain it clearly in the paper.

 Reply: IEEE 3-machine 9-node system is a classic power system model, consisting of 9 nodes and 12 branches.The system is characterized by its moderate size and the inclusion of common power system components, so it is often used in academic research and engineering practice.And it was added on Page 7.

 4. The authors claim that the proposed control method can ensure that the multiple indexes concerned by ultra-complex power systems can be controlled at their expected values under the condition of the uncertainty of system parameters. However, the authors said in the simulation that the system parameters are accurate. This is confusing operation, please make a detailed explanation.

 Reply: In order to verify the effectiveness of MINRAC control law,the control law was applied to IEEE 3-machine 9-nodesystem.The data of line,generator and load were shown in Table 1~Table 4,and the unit of parameters in the table was the standard unit value.Fig.5(a)-(d) showed that both MINRC and MINRAC could maintain the terminal voltage and output active power of the generator at the original given value.Due to the change of network parameters,the power angle of the motor was bound to change to adapt to the new equilibrium point when the terminal voltage and active power remained unchanged. 

Then we can make sure that inn multi-machine power system,for the uncertainty of the generator itself,such as damping coefficient,rotor inertia time constant is inaccurate and generator excitation of electromagnetic disturbance,MINRAC method effect rather than in a single machine infinite system,the mechanism of selection by the output function system of the relationship between indicators such as angle,angular frequency and voltage of the machine as a constraint.By adjusting the uncertain parameter value by adaptive control,these indexes are forced to tend to the given expected value,so as to achieve static tracking.

 5.The authors should add some more recent relevant references in the references section and cite them properly in the paper.

 Reply: We have added and renewed 4 relevant references in the references section and cite them properly in the paper.And the content had been added in “1 INTRODUCTION” on Page 2.

 6. It is recommended that the author reformulate an appropriate title based on the content of the paper.

 Reply: Many thanks for your encouraging comments.In this paper,a Multi-Index Nonlinear Robust Adaptive Control (MINRAC) method was proposed for ultra-complex multi-input multi-output power systems with uncertain parameter factors and external disturbances,and the controller designed by this method has excellent static and dynamic characteristics.After our unanimous discussion,we think that the research content conforms to the content of the article,so we do not change the title of “Research on Robust Adaptive Control of Strong Nonlinear Complex Large Power Grids” .

 7. In conclusion part, more future works and challenges should be recommended.

 Reply: Future works and challenges have been recommended in conclusion part on Page 9.

 8. To improve the quality of the paper, it is recommended to improve the English writing skills.

 Reply: According to your suggestion,we have correct grammatical mistakes of the manuscript.

Comments to the authors of reviewer #2:

 In this work, a Multi-Index Nonlinear Robust Adaptive Control (MINRAC) method has been proposed for ultra-complex multi-input multi-output power systems with uncertain parameter factors and external disturbances. However, in order to further improve the quality of this manuscript, the comments in the uploaded file should be carefully considered.

 Reply: Many thanks for your encouraging comments.We have considered the comments in the uploaded files,and we also have carefully revised the paper.

---

## [Editor Report · Decision Letter 1]

23 Feb 2024

Research on Robust Adaptive Control of Strong Nonlinear Complex Large Power Grids

PONE-D-23-43214R1

Dear Dr.

We’re pleased to inform you that your manuscript has been judged scientifically suitable for publication and will be formally accepted for publication once it meets all outstanding technical requirements.

Kind regards,

Dhanamjayulu C, Ph.D & Post.Doc

Academic Editor

PLOS ONE

Additional Editor Comments (optional):

The authors have revised the properly for reviewers concerns
---

## [Editor Report · Acceptance letter]

24 May 2024

PONE-D-23-43214R1 

PLOS ONE

Dear Dr. Huang, 

I'm pleased to inform you that your manuscript has been deemed suitable for publication in PLOS ONE. Congratulations! Your manuscript is now being handed over to our production team.

Kind regards, 

on behalf of

Dr. Dhanamjayulu C 

Academic Editor

PLOS ONE